# Tissue-selective alternate promoters guide NLRP6 expression

Nathan A Bracey[1,2], Jaye M Platnich[3], Arthur Lau[1,2], Hyunjae Chung[1,2], M Eric Hyndman[4] ⓘ, Justin A MacDonald[5] ⓘ, Justin Chun[1,2] ⓘ, Paul L Beck[1,2], Stephen E Girardin[6], Paul MK Gordon[7], Daniel A Muruve[1,2] ⓘ

The pryin domain (PYD) domain is involved in protein interactions that lead to assembly of immune-sensing complexes such as inflammasomes. The repertoire of PYD-containing genes expressed by a cell type arms tissues with responses against a range of stimuli. The transcriptional regulation of the PYD gene family however is incompletely understood. Alternative promoter utilization was identified as a mechanism regulating the tissue distribution of human PYD gene family members, including NLRP6 that is translationally silenced outside of intestinal tissue. Results show that alternative transcriptional promoters mediate NLRP6 silencing in mice and humans, despite no upstream genomic synteny. Human NLRP6 contains an internal alternative promoter within exon 2 of the PYD, resulting in a truncated mRNA in nonintestinal tissue. In mice, a proximal promoter was used that expanded the 5′ leader sequence restricting nuclear export and abolishing translational efficiency. Nlrp6 was dispensable in disease models targeting the kidney, which expresses noncanonical isoforms. Thus, alternative promoter use is a critical mechanism not just for isoform modulation but for determining expression profile and function of PYD family members.

## Introduction

The innate immune system represents the first line of defense against a multitude of harmful agents within our environment (Akira et al, 2006). Germ line–encoded pattern recognition receptors are proteins expressed in various organ systems that couple detection of injury with effector responses (Liston & Masters, 2017). The repertoire of these sensors expressed by any tissue compartment determines the context by which an inflammatory signal can be generated. The regulation of PRR expression in different cellular populations is therefore an integral component of maintaining system-wide homeostasis.

The ability of PRRs to generate downstream signals is imparted by their modular domain architecture (Pålsson-McDermott & O'Neill, 2007). The pyrin domain (PYD) is a death domain fold superfamily module that contains a 90–amino acid residue motif exclusively found at the amino (N) terminal of various proteins (Bertin & DiStefano, 2000; Chu et al, 2015). When activated, PYD-containing proteins associate through PYD–PYD interactions to regulate assembly of multiprotein complexes that promote inflammation and cell death (Fairbrother et al, 2001). PYD-containing proteins include the NOD-like receptors (NLRs), AIM2-like receptors (ALRs), and regulatory molecules. On the basis of their effector responses, PYD-containing proteins are further subclassified into inflammasome activators, negative regulators, and adaptors (Chu et al, 2015).

The NLRPs are PYD-containing NLR proteins that also include central nucleotide binding (NBD) and C-terminal leucine-rich repeats domains (Martinon & Tschopp, 2005). When stimulated by a wide range of microbial and nonmicrobial signals, they can activate three categories of effector pathways. Firstly, they oligomerize via PYD–PYD interactions with the adaptor apoptosis-associated speck-like protein containing a CARD (ASC) to activate inflammatory caspases, leading to formation of the inflammasome, IL-1-$\beta$/IL18 processing, and pyroptosis (Schroder & Tschopp, 2010; Kayagaki et al, 2015). Secondly, they can directly interact with signal transduction elements to regulate immune signaling (Taxman et al, 2011; Anand et al, 2012). Lastly, they can crosstalk with components of the adaptive immune system through modulation of MHC class I and II expression (Steimle et al, 1993; Meissner et al, 2010). NLRP6 is one unique NLR that may participate in all three pathways (Levy et al, 2017). Several studies have suggested that NLRP6 regulates intestinal IL-18 production downstream of the inflammasome in response to enteric pathogens and microbiota-associated metabolites (Chen et al, 2011; Elinav et al, 2011; Levy et al, 2015). Deletion of *Nlrp6* in mice has also been associated with enhanced MAPK and

---

[1]Department of Medicine, University of Calgary, Calgary, Canada   [2]Snyder Institute for Chronic Disease, University of Calgary, Calgary, Canada   [3]Department of Medicine, University of Alberta, Edmonton, Canada   [4]Department of Surgery, University of Calgary, Calgary, Canada   [5]Department of Biochemistry and Molecular Biology, University of Calgary, Calgary, Canada   [6]Department of Laboratory Medicine and Pathobiology, University of Toronto, Toronto, Canada   [7]Centre for Health Genomics and Informatics, University of Calgary, Calgary, Canada

Correspondence: dmuruve@ucalgary.ca

NFκB signaling following TLR stimulation, suggesting a direct negative regulatory role (Anand et al, 2012). NLRP6 was also shown to regulate a number of interferon-stimulated genes in response to viral RNA through caspase-1–independent interactions (Wang et al, 2015). NLRP6 is therefore emerging as a potential multifunctional PRR capable of eliciting diverse immune responses in various cellular populations.

NLRP6 protein is most highly expressed in the intestinal epithelium where it has been associated with regulating mucosal host–microbiota interactions (Normand et al, 2011; Gremel et al, 2014; Wang et al, 2015). Despite restriction at the protein level, a number of reports in mice have documented *Nlrp6* RNA within broad tissue types including the kidney, liver, lung, lymphocytes, and bone marrow–derived cells (Elinav et al, 2011; Hara et al, 2018; Radulovic et al, 2019). Many PYD-containing genes show similarly diverse expression profiles. For example, NLRP1 and NLRP3 protein are expressed in PBMCs, macrophages, lymphocytes, and dendritic cells (Kummer et al, 2007). cDNA profiling of various tissues has revealed even more diverse patterns of expression though, with many NLRs and regulatory genes showing almost uniform expression across multiple organ systems (Yin et al, 2009). As further complexity, many PYD-containing genes are transcribed as sets of isoform variants that could be regulated through both alternative splicing and differential transcription start sites (TSSs). Few studies however have sought to systematically evaluate the differential contributions of isoform variants to functional responses. It is established that isoforms can have dramatic functional differences in species- and tissue-specific manners, as human, but not mouse, NLRP3 was recently found to undergo splicing within exon 5 of the leucine-rich repeat that gives rise to a nonfunctional isoform (Hoss et al, 2019). Conventional expression profiling studies have used PCR-based techniques against amplicons that cover a range of possible RNA molecules, so our knowledge of which isoforms are functionally relevant remains limited. Moreover, techniques for high-dimensional single-cell RNA sequencing are still evolving the computational capability needed to resolve splicing variants, so isoforms are often aggregated (Arzalluz-Luqueángeles & Conesa, 2018).

Most genes contain multiple TSSs, each reflecting the integration of complex regulatory elements acting in *cis* and *trans* to shape expression patterns (Lenhard et al, 2012). The functional annotation of the mammalian genome 5 (FANTOM5) project mapped TSSs for mammalian genes in various human and mouse cell types through cap analysis of gene expression (CAGE) and single-molecule cDNA sequencing (FANTOM Consortium and the RIKEN PMI and CLST (DGT) et al, 2014). It is now clear that alternative promoters exist for the majority of genes, defined as discrete TSS clusters with varying degrees of tissue-level specificity. Mechanistically, variation in TSS use represents an added layer for tuning gene expression in tissue-specific contexts. Downstream alternate promoters nested internally within a transcript can yield truncated isoforms. Upstream TSS utilization can produce variable leader sequences in the 5′ UTR, which contain upstream ORFs (uORFs) and unfavorable guanine-cytosine (GC) content that impact translation efficiency (Kozak, 1991). Context and tissue-selective 5′UTR variation have been described for the related NOD2 sensor, though little is known regarding the PYD-containing gene family (Rosenstiel et al, 2007).

Here, we used publicly available FANTOM5 CAGE data to map promoters for all PYD-containing genes in various tissues and validated our findings using RNA-Seq. Most PYD genes are broadly expressed using more than one TSS. In human, we identified *NLRP6* as a gene with multiple transcript variants, only one of which codes for full-length translatable protein. Similarly, in mouse, one prominent *Nlrp6* species contains an expanded 5′UTR that abolishes translational efficiency both in vitro and in vivo, resulting in nuclear RNA retention. Both untranslated isoforms represent the dominant RNA species outside of the intestine, suggesting a conserved mechanism for translational gene silencing and tissue-specific expression. We propose that alternative promoters represent a powerful regulatory layer in determining the distribution of many PYD-containing genes across tissue types.

# Results

## Genomic organization and primary structure of the human pyrin domain

Given the central role of the PYD in initiating various innate immune signaling cascades, we looked to profile the tissue distributions and regulatory mechanisms governing the expression for all PYD-containing genes. We retrieved transcript annotations on 21 human PYD genes corresponding to 14 NLRs (*NLRP1-14*), 4 ALRs (*AIM2, PYHIN1, MNDA,* and *IFI16*), and 3 regulators/adaptors (*ASC/PYCARD, PYDC1,* and *MEFV*). The PYD is exclusively expressed at the amino (N) terminus, and its sequence is encoded within a single exon of rank 1 or 2. We first considered any superficial shared relationships in exonic organization and primary nucleotide sequence. PYD domains have a median nucleotide width of 225 nt, though the lengths of the complete exons encoding the domains fall in two groups: one "long" group (*NLRP3* and *ASC* transcripts) with median 1,029 nt and the "short" group (all others) with median 320 nt (Fig 1A, right). We further analyzed amino acid sequences corresponding to the actual PYD domains using multiple sequence alignment and constructed a phylogenetic tree to determine whether there were higher order relationships. Similar to previous reports, three patterns emerged: one cluster was formed by *PYDC1, PYCARD, MEFV, NLRP3, NLRP6,* and *NLRP12,* with the remaining NLRs aligned separately, and the four ALRs formed a third group (Fig 1B) (Fairbrother et al, 2001).

## Characterization of the promoter landscape for all PYD-containing genes

We leveraged publicly available FANTOM5 datasets to computationally explore 5′ centered tissue expression patterns and build promoter maps for human PYD-containing genes (FANTOM Consortium and the RIKEN PMI and CLST (DGT) et al, 2014). CAGE is a high-throughput transcriptome analytical tool that relies on selective retrieval of the 7-methylguanosine–capped 5′ end of Pol II RNA transcripts. The resulting 5′ ends are cleaved, amplified, and sequenced, giving rise to a signal of peaks across the genome that corresponds to 5′ TSSs that can be used to define promoter regions (Kanamori-Katayama

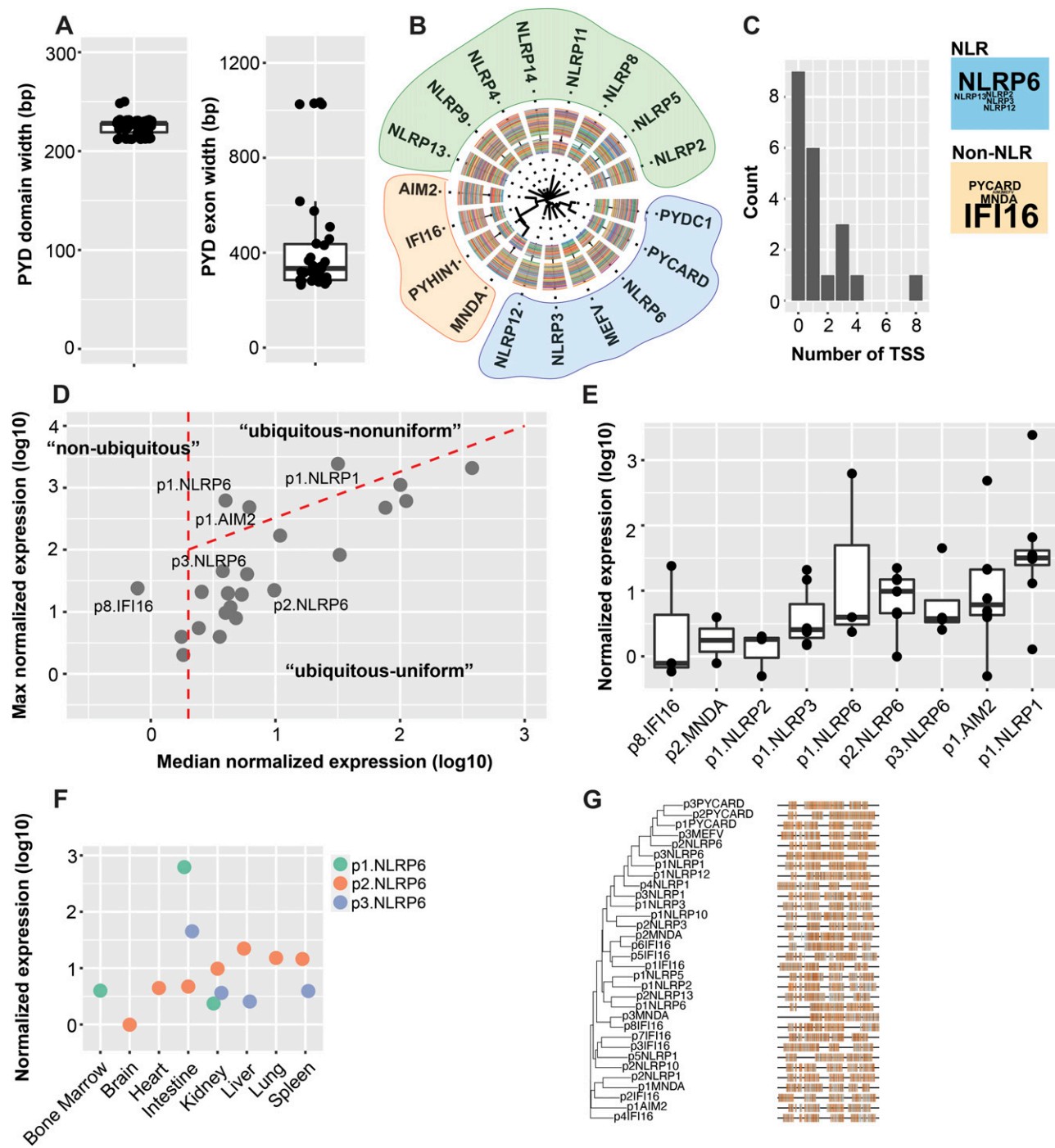

**Figure 1. PYD-containing genes are transcribed by sets of promoters with diverse tissue distributions.**
**(A)** Distribution of Pfam domain (left) and their corresponding exon (right) widths for human PYD-containing genes. **(B)** Phylogenetic tree for all PYD-containing genes aligned by PYD domain. **(C)** Distribution of transcription start site (TSS) consensus cluster counts for all PYD genes in FANTOM5 cap analysis of gene expression data. Word cloud highlights the PYD genes with the most TSS consensus clusters. **(D)** Distribution of normalized maximum and median expression values for PYD-containing gene TSS clusters across tissues extracted from the FANTOM5 database. Dashed red lines indicate boundaries established from initial FANTOM5 analysis of all peak data; to the left of vertical red line are TSS peaks detected with median expression <0.2 tags per million. Below the diagonal line are TSS peaks where maximum <10× median, and above red diagonal are TSS peaks where maximum >10× median. **(E)** Distribution of normalized expression values for select promoter clusters across various tissue types from FANTOM5 data. Note 3 distinct promoters for NLRP6 with diverse tissue distribution profiles. **(F)** Distribution of *NLRP6* promoters across various tissue types in FANTOM5 datasets. **(G)** Phylogenetic tree and alignments for PYD gene promoters (consensus clusters +100 bp upstream).

 **Life Science Alliance**

**Table 1. Characteristics of human NLRP6 transcription start site clusters.**

| Transcription start site | Width (bp) | Exon | Position | GC content | Distribution |
|---|---|---|---|---|---|
| P1.NLRP6 | 54 | Exon 1 | 5'UTR | 0.48 | Non-uniform |
| P2.NLRP6 | 60 | Exon 2 | PYD | 0.60 | Uniform |
| P3.NLRP6 | 155 | Intron 3 | PYD-NACHT | 0.68 | Uniform |

et al, 2011). We explored FANTOM5-pooled sample sets for eight diverse human tissue types: kidney, liver, lung, heart, spleen, intestine, bone marrow, and brain. After normalization, all TSS peaks were spatially clustered by distance into larger transcriptional units, and units between tissue types aggregated to form sets of TSS consensus clusters that reflect putative promoters. We then parsed the data to select for regions corresponding to only PYD-containing genes. Many genes were not expressed under basal conditions in the tissues explored. Not surprisingly, several PYD genes contain multiple promoters, with six genes containing two or more (Fig 1C and Table S1). *NLRP1* and *NLRP6* had the most TSSs of all NLRs with 4 and 3, respectively. For the non-NLRs, *IFI16* was especially diverse with eight possible promoters.

Each promoter had a clear distribution of activity across samples that reflect tissue selective use (Fig 1D and E). To map the promoter landscape for all human PYD genes, we plotted maximum against median scores and separated all promoters into three categories: those where the median score was less than 0.2 tags per million as nonubiquitous, those where the maximum score was greater than 10× the median as ubiquitous and nonuniform, and those where the maximum was less than 10× the median as ubiquitous and uniform. As in the initial FANTOM5 analysis for the human transcriptome, many PYD genes contain alternate promoters that fall in different expression categories (FANTOM Consortium and the RIKEN PMI and CLST (DGT) et al, 2014). For example, P8.IFI16 was selective for bone marrow as nonubiquitous, though the other seven *IFI16* promoters were uniformly distributed across tissues.

To determine whether there were common sequence motifs that could give rise to shared expression profiles, we aligned the DNA sequences for all putative promoters, including +100 bp upstream (Fig 1G). The position of core TSS-adjacent promoter sequences was largely uncorrelated, supporting the individualized tissue-selective distributions noted previously. We used maximum likelihood analysis and bootstrapping to further quantify the relationships (Fig S1). Only two clusters emerged with meaningful alignments: p3.MNDA/p8.IFI16 and p2.MNDA/p6.IFI16. Interestingly, although various promoters for each gene may theoretically encode for the same protein product, they did not cluster together, and instead displayed significantly different sequence profiles. These results reveal the diversity in promoter use for PYD genes across tissue types. They highlight alternative promoters as a possible regulatory mechanism in determining heterogenous tissue-level expression profiles.

### Alternative promoters regulate the tissue distribution for human *NLRP6*

Because there was little correlation between various promoters even for the same PYD genes, we looked in greater detail at their locations and putative transcript products. We focused on *NLRP6*,

given its broad distribution. We found three alternative NLRP6 TSSs: p1, p2, and p3 (Table 1). The p1.NLRP6 site is located within the 5'UTR of exon 1. Interestingly, the other two promoters are localized internally. P2.NLRP6 is within exon 2 in the middle of the PYD sequence, and p3.NLRP6 is within intron 3 between the PYD and NACHT domains. These two transcripts may therefore be presumed not to translate into functional PYD proteins. Despite reports for broad *NLRP6* RNA expression, the use of the p1.NLRP6 site was selective for the intestine. In the kidney, heart, lung, liver, spleen, and brain, the p2.NLRP6 promoter was clearly dominant (Fig 1F).

We verified the tissue-specific *NLRP6* isoforms from the FANTOM5 CAGE datasets experimentally using RNA-Seq on tissue biopsy samples of human kidney and small intestine. Using a deep sequencing count of 266M read pairs per sample, we were able to detect alternate splicing events and TSS use across samples. Indeed, *NLRP6* in small intestine (ileum) used a start site within exon 1 that aligned with the predicted FANTOM5 data (Fig 2A and B). Surprisingly, in the kidney, we detected a truncated *NLRP6* isoform lacking exon 1 corresponding to p2.NLRP6. We looked next at endogenous NLRP6 protein expression in various human tissue types. Similar to previous reports and in contrast to murine tissue, endogenous NLRP6 protein was highly detectable within the small intestine, though not the large intestine. Thus we used small intestine samples for human positive control tissue (Fagerberg et al, 2014). Additionally, a single freeze/thaw cycle disrupted NLRP6 protein signal in the small intestine, so we used only freshly obtained tissues (Fig 2C). Similar to the alternate promoter use, NLRP6 protein was only detectable in small intestine samples (Fig 2C). Therefore, human samples predicted to use p2.NLRP6, such as the kidney yielded no detectable translated protein. The lack of protein signal was not the result of poor specificity or truncated protein variants, as we mapped the epitopes recognized by commercially available human NLRP6 antibodies to the NACHT and PYD–NACHT interface (Fig S2). Together, these results suggest translational repression of human *NLRP6* by alternative promoter use in a tissue-specific context outside of the intestinal epithelium.

### Alternative promoters regulate the tissue distribution for mouse *Nlrp6*

Much of our knowledge regarding PYD gene signaling comes from mouse models with knockout/transgenic approaches. The promoter complexity that we observed in human PYD genes could have species-specific patterns. We therefore went on to fully characterize protein and RNA expression profiles for the NLRs in mouse tissues. In several mouse tissues, *Nlrp6* was the most abundantly expressed NLR at the RNA level under basal conditions, with exon 5–6 amplicons readily detected in the kidney, liver, and intestinal tissue (Fig 3A). However, similar to the human samples, we only detected endogenous Nlrp6 protein in the intestine (both small and large intestines in mouse, Figs 3B and S3). Although Nlrp6 was readily

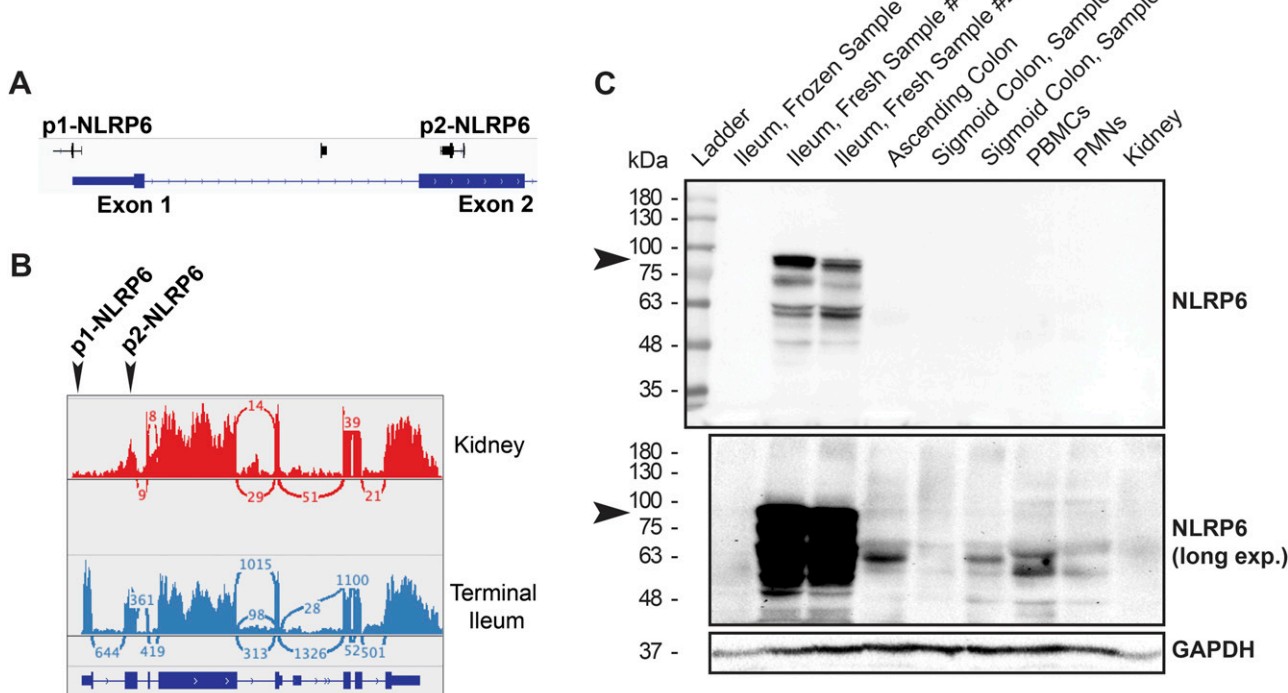

**Figure 2. Human *NLRP6* is regulated by tissue-selective alternate promoters.**
**(A)** Gene-like representation of *NLRP6* transcription start site clusters from FANTOM database. **(B)** Sashimi plot for *NLRP6* showing alternative promoter use of p1.NLRP6 in the representative human small intestine (blue) and p2.NLRP6 in human kidney (red). **(C)** Immunoblot for NLRP6 protein in human fresh and frozen samples (ileum only) for low and high exposures. Arrows indicate predicted NLRP6 size. Source data are available for this figure.
Source data are available for this figure.

detected in the small intestine control tissue, there was no signal evident in kidney lysates prepared with radioimmunoprecipitation assay (RIPA) or urea buffers (Fig 3B). Moreover, endogenous Nlrp6 protein was not induced in kidneys or livers from mice treated systemically with the TLR3 ligand poly (I:C) to induce interferon-dependent gene expression, though it was readily detected in control large intestine colonic tissue (Fig S3). These results suggest that Nlrp6 is not regulated transcriptionally, but rather at the level of protein translation to give rise to tissue-specific expression patterns.

We used RNA-Seq with a sequencing count of 67M single-end reads per sample to further explore the tissue-specific regulation of *Nlrp6*. In the mouse intestine, *Nlrp6* is encoded by eight exons with a 5′UTR of 185 bp in exon 1. Surprisingly, we found that *Nlrp6* in mouse kidney underwent complex alternative splicing at the 5′ end which gives rise to an expanded 5′UTR leader sequence of 1,749 bp (Fig 3C). Splicing occurred between exon 1 of *Nlrp6*, 2 novel upstream intergenic exons and exon 1 of the adjacent upstream gene. *BC024386* is located approximately 14.8 kb proximal to *Nlrp6*, contains three exons, and is annotated as long non-coding RNA with only one 94-bp ORF (Fig 3C and Table 2). We annotated the putative splice sites for the novel *Nlrp6* variant, and they all followed the canonical GT/AG rule (Table 3). We amplified a PCR product using primers directed from exon 1 of *BC024386* to exon 1 of *Nlrp6*. Sanger sequencing confirmed the presence of a splice variant containing exon 1 of *BC024386* and 2 novel intergenic exons leading to a variant *Nlrp6* RNA with an expanded 5′ UTR: Nlrp6Δ5′UTR. Interestingly, *BC024386* was expressed in the

kidney and liver, but not the colon (Figs 3D and S4A). Moreover, Nlrp6Δ5′UTR splicing was generalizable and not the result of genetic inbreeding, as it was also detected in the kidney and liver from mice across various strains (Fig S4B and C). We therefore annotated the proximal exons of *BC024386* as part of the *Nlrp6* genomic locus, representing an alternate promoter for *Nlrp6*.

The previous results suggest that Nlrp6Δ5′UTR is a tissue-selective variant outside of the intestine. Previous reports looking at *Nlrp6* RNA expression have been limited by the use of relative RNA expression against an arbitrary tissue/cell type. We measured various *Nlrp6* amplicons using absolute quantification against standard curves made from sequences of interest. Although the common exon 5–6 region was present in the intestine, kidney, and liver, only the kidney and liver expressed exons 1a–1d and 1e–1f (Fig 3D). We examined *Nlrp6* and BC024386 exon expression in *Nlrp6*[−/−] mice generated by gene targeting and replacement of exons 1–2 with an IRES-bgal-neomycin resistance cassette (Chen et al, 2011). Whereas *Nlrp6*[+/+] littermates expressed abundant Nlrp6Δ5′UTR in the kidney and liver, gene targeting of *Nlrp6* exons 1 and 2 also suppressed expression of BC024386 and the novel intergenic exons within those tissues (Fig 3D).

Taken together, these results show that mouse *Nlrp6* is regulated by tissue-selective alternate promoters that give rise to at least two distinct isoforms: one in the intestine containing the canonical 185-bp 5′UTR and one expressed in the kidney and liver containing a large 1,749-bp 5′UTR.

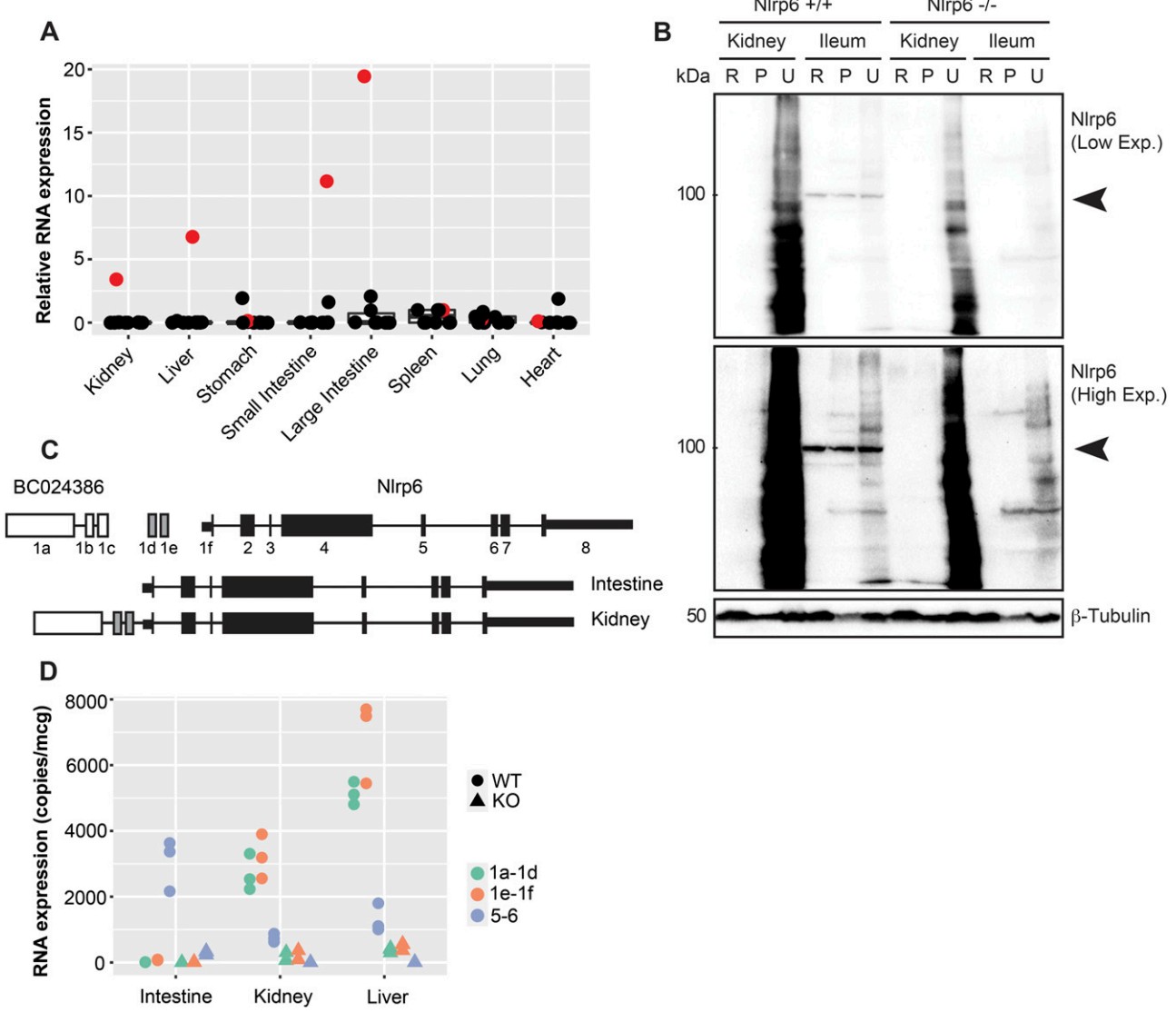

**Figure 3. Murine *Nlrp6* is regulated by tissue-selective alternate promoters.**
**(A)** Tissue distribution of PYD-containing RNA transcripts in organs relative to the spleen (red = *Nlrp6*). **(B)** Immunoblot for Nlrp6 protein in mouse kidney and intestine (R, RIPA; P, insoluble pellet; U, urea). Nlrp6 protein is only detectable in intestine. **(C)** (Top) Genomic organization and reannotation for mouse *Nlrp6* locus. (Bottom) Transcript map for novel *Nlrp6* exons and splicing of tissue-selective 5′UTR leaders in the kidney and intestine. **(D)** Absolute RNA expression of *Nlrp6* amplicons corresponding to different 5′UTR leaders in WT and KO mouse organs. n = 3 biological replicates from littermate mice.
Source data are available for this figure.

## Nlrp6Δ5′UTR is regulated in epithelial cells

To better understand the cellular fate of Nlrp6Δ5′UTR-containing transcripts, we went on to characterize its posttranscriptional regulation. Both protein coding and noncoding RNAs (ncRNAs) can be spliced and polyadenylated (Derrien et al, 2012). To confirm Nlrp6Δ5′UTR splicing, we isolated nuclei from *Nlrp6*[+/+] mouse kidney and measured

expression of each exon–exon junction relative to heteronuclear, unspliced RNA (hnRNA). There was clear enrichment of amplicons overlapping exons 1a–1b and 1a–1d relative to hnRNA (Fig 4A). In contrast, amplicons for exons 1b–1c were not significantly different, suggesting that Nlrp6Δ5′UTR was transcribed as one contiguous primary transcript with subsequent splicing to construct a leader sequence of exons 1a-1d-1e-1f. Moreover, Nlrp6Δ5′UTR was polyadenylated as there

**Table 2. *Nlrp6* and BC024386 characteristics and predicted protein coding scores.**

| RNA | Size (bp) | Longest ORF | Homology to known ORF | CPC score | Predicted class |
|---|---|---|---|---|---|
| BC024386 | 1,673 | 94 | No | −0.363511 | Noncoding (weak) |
| Nlrp6 | 4,438 | 870 | Yes | 8.483620 | Coding |

**Table 3. Nlrp6Δ5′UTR splicing.**

| Intron | Exon | 5′ donor | 3′ acceptor | Exon |
|---|---|---|---|---|
| 1 | 1 AAAG | GTTAGTGCTC | ATTTTTATCTTTCAG | 2 CTTC |
| 2 | 2 TGAT | GTGAGACCTA | TCCCGGTGTCTGCAG | 3 AGGC |
| 3 | 3 TTCT | GTGAGTGCGT | TATCCCTGCCCACAG | 4 GCCC |

was clear enrichment of Nlrp6Δ5′UTR in samples prepared with cDNA primed from oligodT compared to random hexamers (Fig 4B).

In the intestine, *Nlrp6* RNA expression has been found primarily in enterocytes and colonic goblet cells (Chen et al, 2011; Elinav et al, 2011; Normand et al, 2011). It has also been described in various circulating immune cell populations including macrophages and lymphocytes measured by relative RNA expression (Hara et al, 2018; Radulovic et al, 2019). To establish which cells within the kidney express Nlrp6Δ5′UTR, we fractionated fresh single-cell kidney preparations across a density gradient and sorted the samples by flow cytometry before absolute quantification by real-time PCR. The "parenchymal" layer from density gradient separation alone retained high Nlrp6Δ5′UTR expression (Fig 4C). Further separation of this population revealed dominant expression within E-cadherin+/CD45− epithelial cells. This is

consistent with publicly available single-cell RNA-Seq kidney atlases, which support *Nlrp6* RNA expression exclusively within epithelial cell populations (Wu et al, 2018). Crude leukocyte populations containing macrophages (CD45+/F4/80+), neutrophils (CD45+/Ly6G+), T lymphocytes (CD45+/CD3+), and B lymphocytes (CD45+/IgM+) expressed very little NLRP6Δ5′UTR. Interestingly, mouse kidney tubular epithelial cells in two-dimensional culture lost expression of Nlrp6Δ5′UTR after a single generation (Fig 4C, TECp0), and kidney mesangial cells were below the limit of detection. Some reports have suggested that mouse BMDMs can form functional Nlrp6 inflammasomes, although others have found that FLAG-tagged Nlrp6 protein was restricted to the intestinal tissue in mice (Wang et al, 2015; Hara et al, 2018). Consistent with the latter, we did not detect any *Nlrp6* isoforms in BMDM either in resting states or following LPS stimulation (Fig 4C).

## Nlrp6Δ5′UTR is translationally silenced and retained in the nucleus in the kidney

Alternate promoters that give rise to variable leader sequences can impact protein translation through several different mechanisms. First, splicing of new genetic material can simply disrupt the ORF.

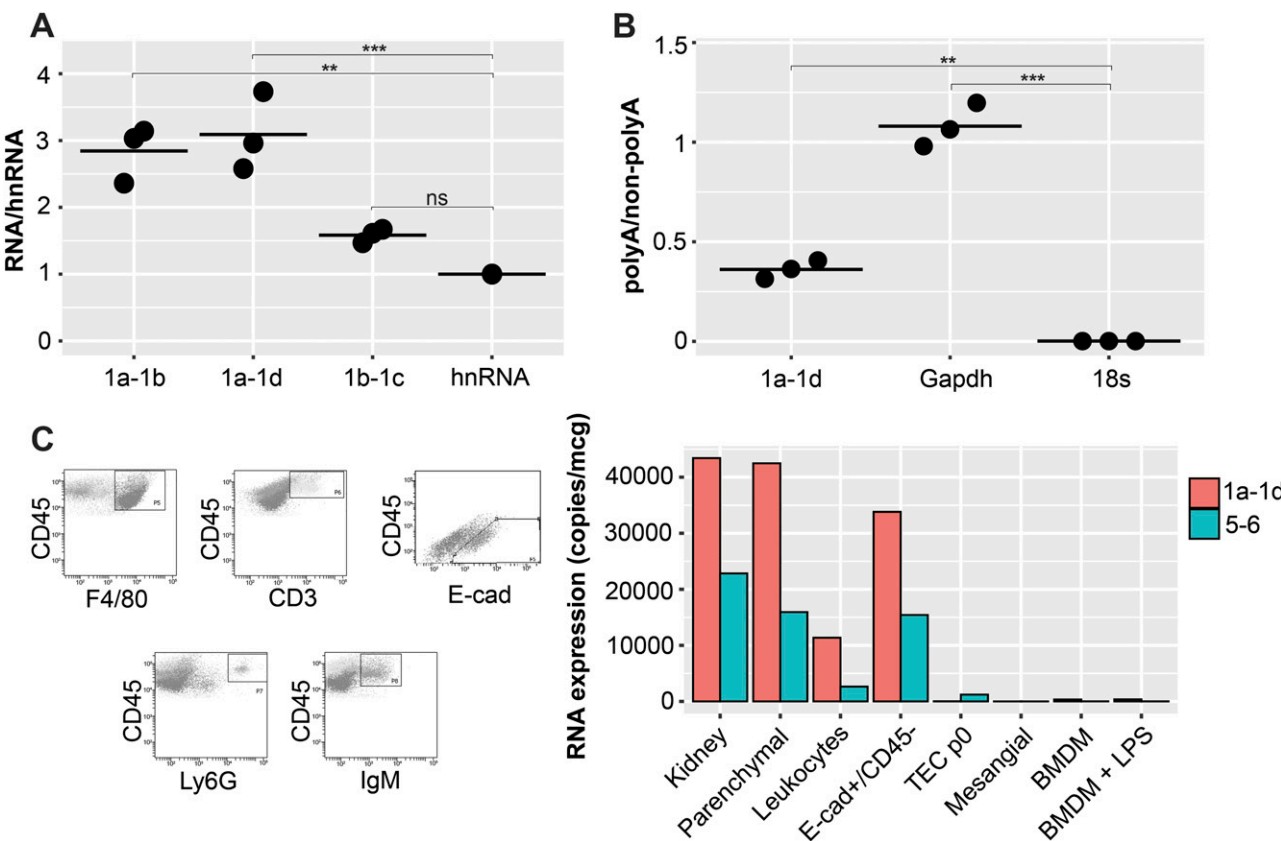

**Figure 4. Nlrp6Δ5′UTR variant is spliced and polyadenylated in kidney epithelial cells.**
**(A)** Nlrp6Δ5′UTR RNA expression relative to *Nlrp6* hnRNA in nuclei isolated from whole kidney. **(B)** Nlrp6Δ5′UTR RNA expression in polyA versus non-polyA whole-cell kidney RNA preparations. n = 3 biological replicates, *P*-values *0.05, **0.01, ***0.001, ****0.0001 by ANOVA with Tukey's multiple comparison. **(C)** Density gradient separation and flow sorting of kidney cells. (Left) Hierarchical gating for macrophages (CD45[+] F4/80[+]), neutrophils (CD45+Ly6G+), T lymphocytes (CD45[+] CD3[+]), B lymphocytes (CD45[+] IgM+), and epithelial cells (CD45− E-cadherin+). (Right) Absolute RNA expression for *Nlrp6* amplicons in various cell populations. TEC, tubular epithelial cells in 2D culture; BMDM, bone marrow–derived macrophages in 2D culture; mesangial cells in 2D culture. Representative experiment from n = 6 pooled kidneys. hnRNA, heteronuclear RNA.

**Table 4.** *Nlrp6* UTR characteristics.

| 5′UTR variant | Length (bp) | GC content | ORF count | uORF length (aa) |
|---|---|---|---|---|
| Exon 1f | 185 | 0.53 | 1 | 31 |
| Exon 1a-d-e-f | 1,749 | 0.51 | 5 | 101, 79, 62, 31, 31 |

Alternatively, UTRs can impact cap-dependent ribosomal binding elements of translation through the primary structure with changes in the GC content and insertion of uORFs, or the secondary structure through hairpins and pseudoknots (Kozak, 1991; Sonenberg, 1993; Wang et al, 1999). We compared the UTR of Nlrp6Δ5′UTR with the canonical intestinal transcript (Table 4). Nlrp6Δ5′UTR is 1,749 nt, preserves the ORF, and contains five uORFs with only slightly lower GC content than the 1f-UTR.

It is likely that the addition of ~1,500 nucleotides to a leader sequence would significantly impact protein translation even with preservation of the ORF. To assess whether Nlrp6Δ5′UTR is actively translated in vivo, we used polysome profiling. Polysome preparations were made from whole mouse kidney and intestine, size-fractionated along a sucrose gradient, and the purified RNA retrieved from eluted samples was used for cDNA synthesis and quantitative real-time PCR (Fig 5A). As expected, *Gapdh* RNA was enriched in polysome-containing fractions in both kidney and intestine, reflecting an actively translated RNA species in both tissue types

(Fig 5B). In contrast, *Nlrp6* detected by a shared exon 5–6 amplicon was only associated with polysomes in the intestine. In the kidney, *Nlrp6* RNA was most abundant in fractions 1–8, reflecting free untranslated RNA.

To directly compare the translational efficiency of the Nlrp6Δ5′UTR and canonical *Nlrp6* leader sequences, we in vitro transcribed chimeric RNAs containing each *Nlrp6* UTR upstream of luciferase under an SP6 promoter (Fig 5C). Each RNA chimera was 5′ capped and polyA tailed. Translational efficiency was then assessed by measuring luciferase activity relative to luciferase control RNA with no leader. As expected, the Nlrp6Δ5′UTR leader completely abolished translational efficiency, with no signal detected at both 30 and 90 min (Fig 5D). In comparison, the canonical *Nlrp6* leader sequence was actively translated.

The cellular fate of untranslated RNAs is diverse—some messages are exported from the nucleus where they can interact with cytoplasmic molecules, whereas others are retained in the nucleus and participate in various signaling pathways (Quinn & Chang,

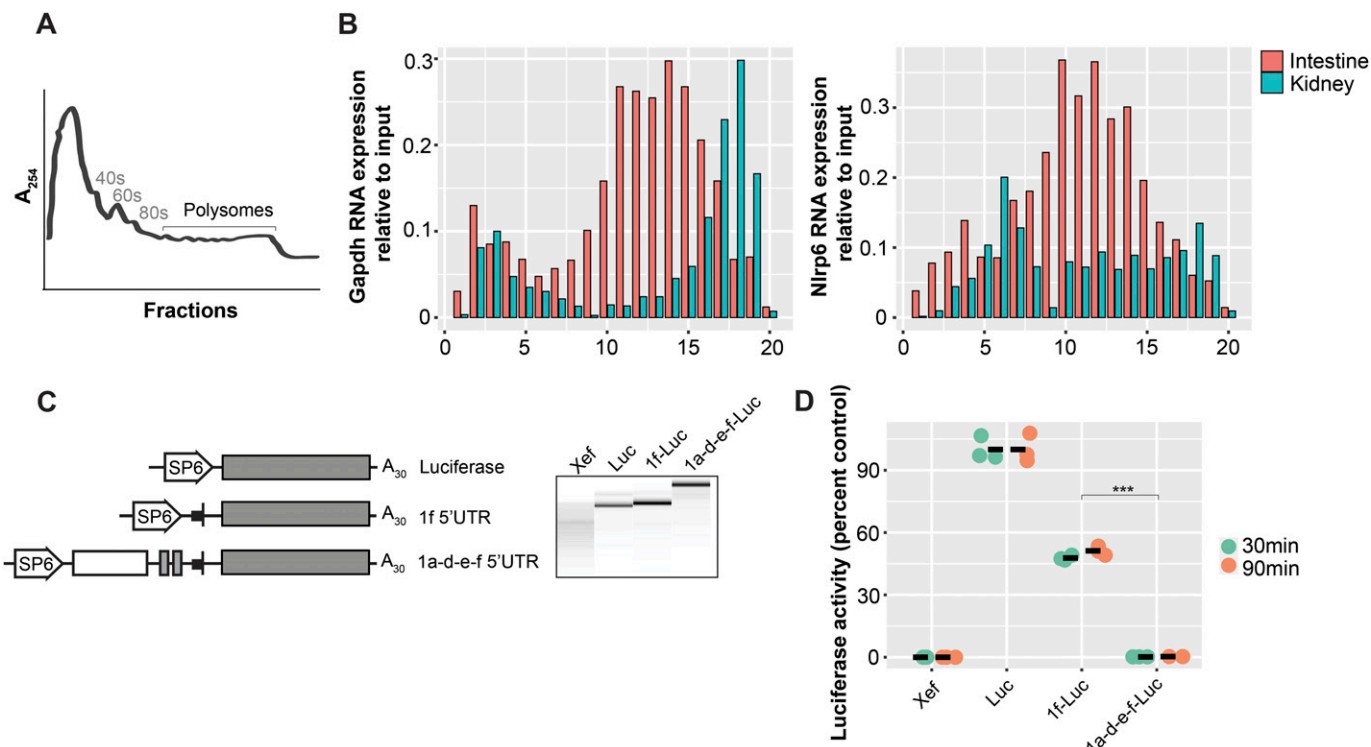

**Figure 5. Nlrp6Δ5′UTR isoform has reduced translational efficiency.**
**(A)** Representative tracing for polyribosome profiling of mouse kidney tissue. **(B)** *Nlrp6* RNA expression in polysome fractions from mouse kidney and intestine. *Gapdh* is comparison for an actively translating mRNA. **(C)** Chimeric RNA constructs and transcripts (right panel) used for in vitro translation. All were 5′ capped and contained the SP6 promoter, designated leader sequences, luciferase reporter, and poly A tail. **(D)** In vitro translation of capped and tailed *Nlrp6* 5′ leader RNA constructs. *Xef* RNA is negative control. Results are expressed as percent of luciferase control containing no leader sequence, n = 3 biological replicates translated in separate reactions. *P*-values *0.05, **0.01, ***0.001, and ****0.0001 by ANOVA with Tukey's multiple comparisons test.

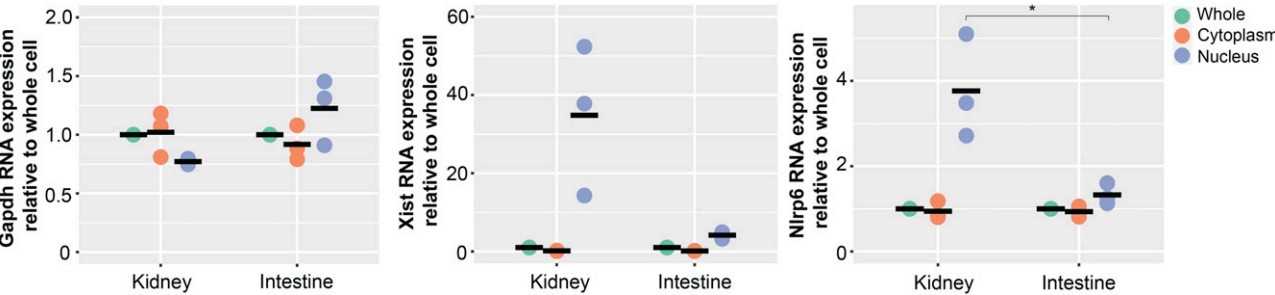

**Figure 6. Nlrp6∆5′UTR is associated with tissue-selective *Nlrp6* nuclear retention.**
Nuclear/cytoplasmic fractionation and absolute *Nlrp6* RNA expression in mouse kidney and intestine. *Gapdh* serves as control for cytoplasmic RNA, and *Xist* for nuclear RNA. Note the scale for *Xist* as 10-fold greater than the others reflecting high nuclear concentration in both kidney and intestine. *$P < 0.05$ by ANOVA with Tukey's multiple comparison test, n = 3 biological replicates from littermate mice.

2016). Curiously, mouse *Nlrp6* RNA was previously identified to be enriched in the nucleus in a study specifically exploring liver tissue, although the mechanism and relevance remained unclear (Bahar Halpern et al, 2015). To determine whether different isoforms of mouse *Nlrp6* are spatially distributed and contribute to nuclear accumulation, we performed nuclear cytoplasmic fractionation of mouse kidney and intestinal tissue. As expected, *Gapdh* was evenly distributed in the cytoplasm and nuclei, and the nuclear-specific long non-coding RNA *Xist* was strictly nuclear in both kidney and intestine, reflecting pure fractions (Fig 6) (Clemson et al, 1996). Interestingly, only the Nlrp6∆5′UTR RNA isoform from the kidney was enriched in the nucleus. In stark contrast, the canonical *Nlrp6* isoform in the intestine was distributed in a similar profile to *Gapdh* between the nucleus and cytoplasm, consistent with a translating mRNA.

Taken together, the aforementioned results confirm that only the canonical intestine *Nlrp6* mRNA isoform is actively exported to the cytoplasm for translation. In contrast, the Nlrp6∆5′UTR variant found in the kidney was associated with impaired translational efficiency and nuclear accumulation.

### *Nlrp6* is dispensable in the kidney

The observation that the Nlrp6∆5′UTR isoform silences protein expression outside of the intestine raises the question of whether the RNA molecule itself still has a functional role as a ncRNA. To address this, we performed RNA-Seq on the kidney tissue isolated from *Nlrp6⁺/⁺* and *Nlrp6⁻/⁻* littermate mice and explored whole transcriptomes for differential gene expression under baseline conditions. The transcriptomes were nearly identical between *Nlrp6⁺/⁺* and *Nlrp6⁻/⁻* mice at baseline, with only two genes identified that were significantly up-regulated: *Ifitm2* with $\log_2$fold change of 1.6 and adjusted *P*-value $5.99 \times 10^{-12}$; and *Pgghg* with $\log_2$fold change of 1.8 and adjusted *P*-value of $2.14 \times 10^{-33}$ (Fig 7A and B). Importantly, both *Pgghg* and *Ifitm2* are located on chromosome 7 immediately upstream of *Nlrp6* by 12.4 and 19.7 kb, respectively. This suggests likely nonspecific *cis*-mediated changes secondary to *Nlrp6* gene targeting and subsequent local chromatin alterations, rather than biologically significant gene regulation.

We considered whether ncRNA signaling effects could be disease- or injury-dependent. To this end, we performed experimental

unilateral ureteric obstruction (UUO) on *Nlrp6⁺/⁺* and *Nlrp6⁻/⁻* mice. UUO is an epithelial-centered injury model that leads to kidney tubulointerstitial inflammation, cell death, and fibrosis. Consistent with the differential gene expression results, there was no difference in histological scoring of CD11b+ cellular infiltrate or markers of fibrosis between *Nlrp6⁺/⁺* and *Nlrp6⁻/⁻* mice at 7 and 14 d (Fig 7C–F). Furthermore, *Nlrp6* RNA was not induced in response to the injury. Amplicons directed against both exons 5–6 and 1a–1d decreased substantially in ligated kidneys compared with contralateral controls, suggestive of nonspecific loss of tubular epithelial cell mass (Fig 7G). We also used a glomerular kidney injury model to further assess the role of Nlrp6 within the kidney. Infusion of sheep-derived anti-glomerular basement membrane (anti-GBM) serum results in a primary glomerular injury with crescent formation, secondary tubular cell injury, and albuminuria (Mesnard et al, 2009). Both *Nlrp6⁺/⁺* and *Nlrp6⁻/⁻* mice developed similar histological injuries by 10 d (Fig 7H). As in UUO, there were no phenotypic differences. Both *Nlrp6⁺/⁺* and *Nlrp6⁻/⁻* mice had similar degrees of albuminuria, and there were no differences in the number of crescents found on histology (Fig 7I and J). Neither UUO nor NTS resulted in any detectable Nlrp6 protein in the kidney (Fig S5). Overall, these in vivo results suggest that Nlrp6 is dispensable within the kidney and that alternative *Nlrp6* promoter utilization operates primarily as a means of tissue-selective translational gene silencing.

## Discussion

It has long been recognized that alternate promoters regulate gene expression. For example, the human dystrophin gene contains at least five promoters used in tissue- and development-specific patterns (Ahn & Kunkel, 1993). Human *NOS1* is especially complex with nine exon 1 leader isoforms in various tissues, each imparting unique changes to translational efficiency (Wang et al, 1999). Although the pathways by which variant promoters regulate protein expression are known, the magnitude of impact has only recently become apparent with efforts to fully characterize mammalian transcriptomes across tissue types. Indeed, the vast majority of genes contain alternate promoters and display cell type–restricted expression profiles, with only a very small minority

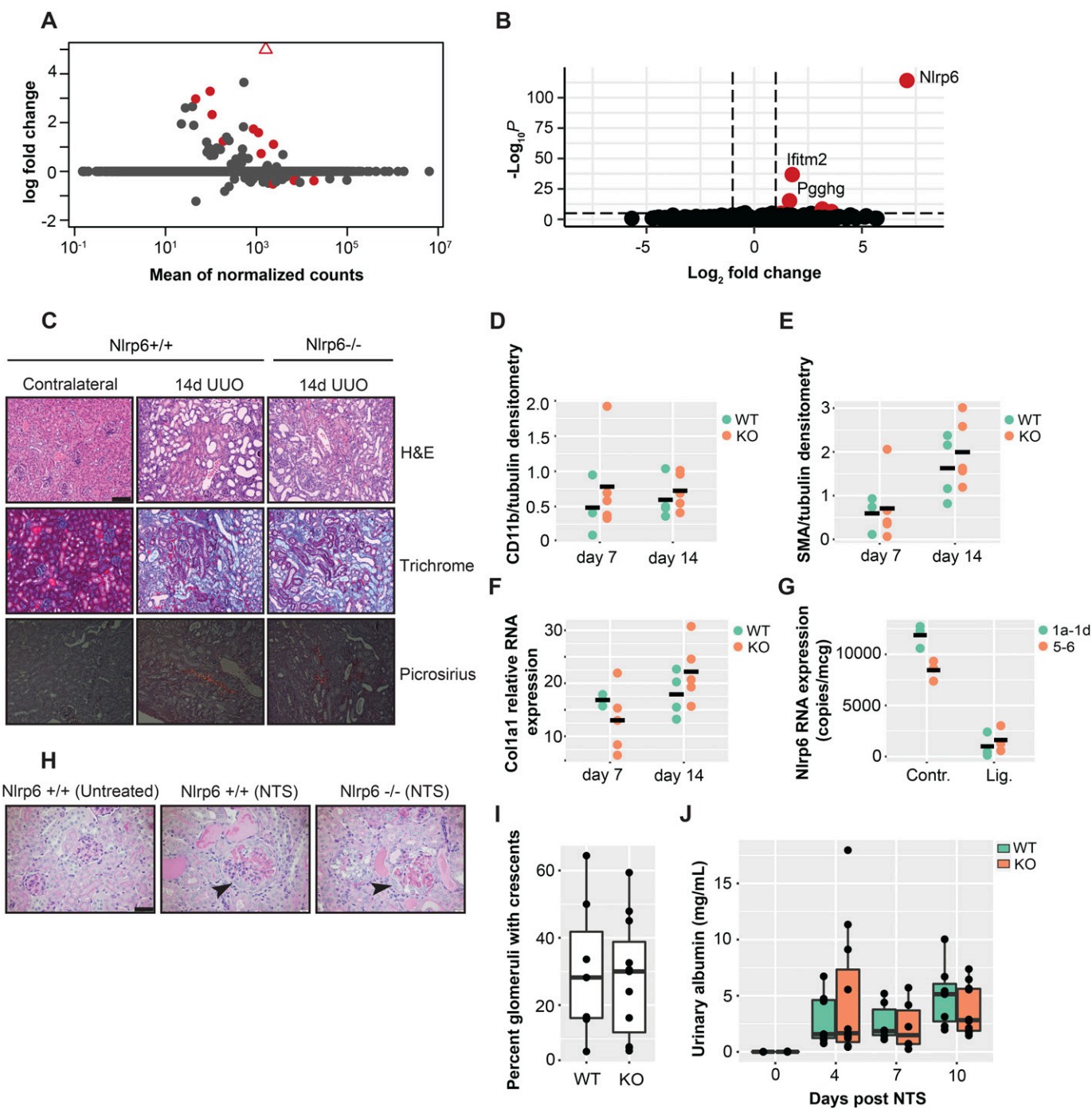

**Figure 7. *Nlrp6* is dispensable in kidney epithelium.**
**(A)** MA plot showing differential gene expression in the kidney from *Nlrp6$^{+/+}$* and *Nlrp6$^{-/-}$* littermates. Red signifies adjusted *P*-values < 0.1, triangle is *Nlrp6*. **(B)** Volcano plot highlighting similarities between *Nlrp6$^{+/+}$* and *Nlrp6$^{-/-}$* kidney gene expression. All represent n = 3 littermate mice per group. **(C)** Representative histological sections from *Nlrp6$^{+/+}$* and *Nlrp6$^{-/-}$* mice at 14 d showing H&E (top), Trichrome (middle), and Picrosirius Red for contralateral controls and unilateral ureteric obstruction (UUO) kidneys. Bar is 80 $\mu$m. **(D, E)** CD11b and αSMA quantitative densitometry of protein expression by immunoblotting. **(F)** *Col1a1* relative RNA expression in *Nlrp6$^{+/+}$* and *Nlrp6$^{-/-}$* kidneys following UUO. **(G)** Absolute *Nlrp6* RNA expression in contralateral control and ligated *Nlrp6$^{+/+}$* mouse kidneys at day 14 UUO. **(H)** Representative PAS-stained kidney sections from *Nlrp6$^{+/+}$* and *Nlrp6$^{-/-}$* mice at 10 d following nephrotoxic serum (NTS). Black arrows point to glomeruli with crescents. Bar is 40 $\mu$m. **(I)** Percent of crescentic glomeruli for NTS mice at 10 d. **(J)** Urinary albumin from mice following NTS injury. All represent n = 3–7 mice per group for F1 littermate mice.

truly aligned with "housekeeping" activities (FANTOM Consortium and the RIKEN PMI and CLST (DGT) et al, 2014). When taken together, co-expression clustering of all mammalian promoters has revealed

that a dominant component of the genome is dedicated to the immune system. Different organs and tissues face very different threats. One might therefore expect heterogenous expression of

genes controlling polarizing events such as inflammatory cytokine release and programmed cell death. It would be surprising if innate immune sensors were evenly distributed without fine regulation; compartmentalization is critical for immune protection while, at the same time, preventing collateral damage.

In contrast to prior observations that the PYD gene family members are broadly expressed, our analysis is more consistent with a mosaic pattern with different cell types expressing different RNA isoforms that can give rise to different levels of protein products under basal states. Such a complex system would arm tissues with tunable way to regulate expression at the level of translation. Arguably, this could represent a more time- and energy-efficient way to express PYD genes strictly on an as-needed basis. For example, epithelial cells in different organ systems possess functional differences while retaining a common cellular phenotype. The use of alternate promoters shifts the burden of managing genetic regulation from the level of DNA to RNA, allowing greater flexibility in determining which specialized transcripts are ultimately "on" or "off" in common cell types comprising different tissues.

With regard to NLRP6, our analysis reveals broad translational silencing in various tissue types. Although BC024386 and the Nlrp6 proximal promoter region are not conserved between mice and humans, the functional paradigm of alternate promoters giving rise to different isoforms in the intestine compared with other tissue types is similar. The epithelial cells of the intestine face unique environmental challenges that are not encountered by epithelial cells of the kidney and liver, which are typically sterile (Peterson & Artis, 2014). Therefore, our findings continue to support a selective role for NLRP6 possibly in regulating or responding to the complex intestinal microbiota. These needs could have imparted evolutionary selective pressure to continue suppressing NLRP6 protein expression outside of the intestine while still maintaining transcription.

The absence of Nlrp6 protein in the kidney and liver does not entirely preclude a functional biological role. Prior animal studies have found that *Nlrp6* deletion results in more severe injury in a chemical-induced acute kidney injury mouse model, although littermate controls were not included (Valiño-Rivas et al, 2020). Previous studies for *Nlrp6* in regulating the intestinal microbiota and response to colitis models have shown significant heterogeneity depending on the use of littermate controls (Lemire et al, 2017; Mamantopoulos et al, 2017). Moreover, the aforementioned Nlrp6 expression in the kidney through immunoblotting and histochemistry did not use knockout animals as negative controls, raising questions of specificity. Lastly, the discrepancy between our findings could relate to the difference in disease models—both UUO and NTS are chronic models of kidney injury, whereas the prior role for Nlrp6 was proposed to regulate acute kidney injury. It is less likely that strain-specific effects are at play, given the generalizability of Nlrp6Δ5'UTR expression across strains of mice. Aside from the two proximal genes (*Ifitm2* and *Pgghg*), we did not detect any meaningful differentially expressed genes in kidneys from littermate *Nlrp6*$^{+/+}$ and *Nlrp6*$^{-/-}$ animals at baseline nor were there distinguishable phenotypes in either UUO or NTS disease models targeting the tubular epithelium and the glomerular compartment. Interestingly, IFITM2—one of the Nlrp6-proximal genes that was differentially expressed—is an interferon-inducible protein that prevents viral entry to the cytoplasm and is expressed in BMDM (Wrensch et al, 2015). It remains to be explored whether the phenotypes previously observed in BMDM and kidney injury attributed to Nlrp6 could in fact relate to off-target effects from gene targeting. Taken together, though, our results suggest that tissue-selective promoter utilization for NLRP6 functions solely as a means for translational silencing at baseline, a process that does not seem to be affected during organ injury or by exogenous stimuli used in this study. A regulatory role for noncoding NLRP6 RNA species in other tissues or disease contexts however cannot be entirely ruled out.

The characterization of the Nlrp6Δ5'UTR isoform as dominant in the kidney and liver reveals the mechanism for nuclear retention of *Nlrp6* RNA that has been observed in previous studies (Bahar Halpern et al, 2015). We have additionally uncovered a novel mechanism whereby Nlrp6 is silenced through alternative splicing to generate a nontranslatable isoform. Although it is clear that this process silences Nlrp6 expression, we did not extensively explore whether there were other physiological or pathophysiological circumstances leading to shifting promoter use within the same tissue leading to context-dependent translational release. This work begins to define the PYD gene promoter landscape under baseline conditions. Future studies should further examine basal and injury-induced regulation of promoter use at a systems level and their impact on isoform and protein expression during innate immune signaling.

# Materials and Methods

### PYD gene family domain and exon analysis

PYD-containing genes were identified using the ensembldb package in Bioconductor (Rainer et al, 2019). All transcripts on UCSC GRCh38 were subset for domains annotated on the Pfam database with protein domain ID PF02758 (PYRIN). Genomic coordinates for both the PYD domains and their exons were extracted by transcript for analysis.

### FANTOM5 CAGE TSS clustering

FANTOM5 CAGE data were accessed and analyzed using the CAGEr package in Bioconductor (Haberle et al, 2015). The FANTOM5 database was queried, and we compiled CAGE data corresponding to human kidney, intestine, heart, lung, bone marrow, liver, brain, and spleen samples. Raw peaks were normalized against a power law distribution with $\alpha$ 1.14 and T = $10^7$. TSSs were clustered in a 20-bp framework, and consensus clusters between all tissues were calculated using a tags per million threshold of two and a maximum distance of 100 bp for the 0.1–0.9 quantiles. We parsed the dataset to select only PYD-containing genes using ensembldb and analyzed clusters overlapping PYD gene coordinates on UCSC GRCh37. We compiled each consensus cluster and the corresponding scores to visualize the distributions according to thresholds defined by FANTOM5 analysis of all TSS cluster (FANTOM Consortium and the RIKEN PMI and CLST (DGT) et al, 2014). Signal peaks were exported and visualized on the Integrative Genomics Viewer (IGV) 2.4 browser.

## Multiple sequence alignment and phylogenetic tree analysis

Annotated genomic sequences for both the PYD domains and TSS clusters were translated to amino acid sequences and compiled into FASTA formats. These sequences were then aligned using MUSCLE, and the phangorn package dist.ml function was used to construct distance matrices and trees (Edgar, 2004; Schliep et al, 2017). Parsimony scores for each model were compared, and we selected neighborhood joining clustering models. We separately used maximum likelihood methods and bootstrapping to verify results and provide statistical analysis (Douady et al, 2003). Trees and alignments were visualized using the ggtree package in Bioconductor (Yu et al, 2018).

## Human tissue samples

Human intestinal tissues were obtained during colonoscopy performed as part of colon cancer screening. A minimum of six biopsies taken for each site were assessed. Human kidney cortex tissue was obtained from the normal margins of kidney specimens from patients undergoing a surgically indicated nephrectomy. Fresh human tissues were immediately rinsed twice in cold PBS, cleaned of adventitia, and kept on ice for parallel processing. RNA was immediately extracted from 100 mg fragments by Solution D method.

## Protein immunoblotting

The tissue was rinsed in saline and processed in RIPA or urea buffers. Protein samples were separated on SDS–PAGE gels under reducing conditions and transferred to nitrocellulose membranes, blocked for 1 h in 5% BSA or milk proteins diluted in PBS or TBS containing 0.5% or 0.1% Tween 20, respectively. Following blocking, membranes were incubated at 4°C overnight with the following antibodies: rabbit anti-mouse Nlrp6 (E-20; Santa Cruz), mouse anti-human NLRP6 (Clint-1; Adipogen), rabbit anti-human GAPDH (Cell Signaling), mouse anti-human tubulin (Sigma-Aldrich), mouse anti-human NLRP6 (R&D Systems), rabbit anti-GFP (Thermo Fisher Scientific), rabbit anti-mouse CD11b (Abcam), and mouse anti-mouse SMA (Clone 1A4; Sigma-Aldrich). Membranes were incubated for 1 h at room temperature with the secondary antibody at 1:5,000 in blocking buffer and visualized using ECL Western blotting detection reagents (Amersham, GE Healthcare). Images were captured using a ChemiDoc MP imaging device (Bio-Rad Laboratories).

## RNA isolation and cDNA preparation

Whole cell RNA was extracted from 50 mg fresh tissue samples or cultured cells (10 cm plates) using guanidinium thiocyanate/phenol–chloroform extraction by using the Solution D method (Chomczynski & Sacchi, 2006). Samples were treated with DNase I (5U, 15 min at room temperature). cDNA synthesis was carried out using Superscript II RT according to the manufacturers' protocol with 100 ng of whole cell RNA and random hexamers. Final cDNA products were diluted to 100 $\mu$l before use.

## Quantitative real-time PCR

qRT-PCR was performed in 10 $\mu$L volumes using SYBR Green detection (Bio-Rad) on a CFX96 Touch sequence detection system (Bio-Rad Laboratories). For comparative Ct, fold change was normalized to Gapdh. For absolute quantification, standard curves were made by 10-fold dilution using gene fragments containing target sequences (Integrated DNA Technologies). Table 5 shows all oligosequences used for amplification.

## RNA sequencing

Whole cell RNA from littermate Nlrp6^+/+ and Nlrp6^−/− mouse kidneys in triplicate or human samples was used. RNA samples were assessed using a Qubit fluorimeter with both double stranded DNA- and RNA-specific fluorescent dyes. RNA integrity (RIN score) was determined by using an Agilent TapeStation 2200 instrument. Ribosomal RNA depletion of the total RNA samples was performed using the NEB ribosomal RNA depletion kit (E6350) as per the manufacturer's protocols. RNA samples were converted into Illumina compatible cDNA sequencing libraries using NEB Ultra II Directional RNA Library Prep kits for Illumina (E7760) and NEB Index primers as per the manufacturer's protocols. Before pooling and sequencing, each library was quantitated by qRT-PCR, in triplicate, using a Kapa Biosystems #KK4835 (#07960204001; Roche) Library Quantification Kit for Illumina. qPCR was performed on an Applied Biosystems StepOne Plus instrument. Equal amounts of each library were combined into a single pool, denatured, and diluted as per Illumina's recommendations. The pool was then immediately sequenced on either an Illumina NextSeq 500 sequencer using a high-output 2 × 75 cycle sequencing kit or on an Illumina NovaSeq 6000 sequencer with a 2 × 50 bp sequencing kit using the manufacturer's protocol.

Spliced paired-end read genomic alignment was performed using a Dragen v3.5.7 (Illumina Inc.) in two-pass RNA mapping mode against GRCh38 for human and using minimap (v 2.17) in short genomic paired-end reads mode against GRCm38 (Li, 2018).

Differential gene expression was carried out using the DESeq2 package in Bioconductor (Love et al, 2014). After genomic alignment, unnormalized count matrixes were loaded to a DESeqDataSet and differential expression was quantified using the DESeq command. Shrunken log$_2$fold changes were computed using the apeglm package and the lfcShrink command (Zhu et al, 2019).

## Flow cytometry and cell sorting

Whole mouse kidneys and intestine were perfused with cold saline, rinsed, and resected. Single-cell suspensions were generated in Multi-Tissue Dissociation Kit-2 on a gentleMACS Octo Dissociator with Heaters (Miltenyi Biotec) using the 37MTDK-2 setting. Suspensions were then filtered through 40-$\mu$m nylon strainers, rinsed in saline, and fractionated across a 9-ml 25–40–60% Percoll (GE Healthcare) density gradient. The lower layer corresponded to "parenchymal" resident cells and the upper layer to immune cell populations and mesangial cells. Separate fractions were then incubated in Fc block at 1 $\mu$g/10$^6$ cells for 5 min at room temperature, followed by primary antibody labeling with anti-CD45

**Table 5.  Oligonucleotide sequences.**

| Amplicon | Forward (5′-3′) | Reverse (5′-3′) |
|---|---|---|
| 18s | TCAGCCACCCGAGATTGAGC | GTGCAGCCCCGGACATCTAA |
| Gapdh | GGTGCTGAGTATGTCGTGGA | GGCGGAGATGATGACCCTTT |
| Xist | GCTTGAACTACTGCTCCTCCG | GGCAATCCTTCTTCTTGAGGCA |
| Nlrp6 1a-1f | ATCCTGAATGCCCAGCGGTGCC | CTCATTCTGGGTGTGAGGGTGT |
| Nlrp6 1a-1d | CGTGCATCAGACGCTCTCTA | GGCAGGAAGGAATTTGAGGC |
| Nlrp6 exon 1e-1f | CTGGAACTGGACTTACGGGT | GGGCAGAAGGTTGGAGAGAT |
| Nlrp6 exon 5-6 | GTGAGACAATGACTACCCCGAAAT | GTCTCGGCAAACTGCATCAG |
| BC024386 exon 1 | CGTGCATCAGACGCTCTCTA | |
| BC024386 exon 1-2 | | TCGCACTCACTAAGCCATGG |
| BC024386 exon 1-intron 1 | | TGCCATTGTTTTGCAGATTTGG |
| BC024386 exon 1-intron 2-exon 3 (hnRNA) | GCTTGGACACGCACAGAATC | CCACTTCCTCAGCCCTGTATG |
| Nlrp1b | GACTTTGTGGCTTGTTGAATG | CATTTAGCTGCAGGTCTAGCTCTCT |
| Nlrp2 | CCCTGCAAATGCTTAGATTGAA | GGTCACTGCTGATTCTCAGTTG |
| Nlrp3 | AGAGCCTACAGTTGGGTGAAATG | CCACGCCTACCAGGAAATCTC |
| Nlrp4a | TTGCTGCCCACTGCTTAAAAC | CAGCCTTTCCATATAGCTGTGTTC |
| Nlrp5 | GGCCAAAAATAGAGTGGGAGTAAAA | GGCCACAGTTGTCCAGTATCAAC |
| Nlrp9a | GTTATGGTTGCCTGGTTGCTATTT | TTATTGTTGCCAAGTTTCAGGGTCTTT |
| Nlrp10 | AACAGGGTCTCAGGCAGTCAAG | ATCCACACCTGGGAGATGCA |
| Nlrp12 | AGCGTGGTATATCCCTCGAAGA | CCCTGAGCATCATGGAAAGAA |
| Nlrp14 | GAGAGACTGGCCTTAGCAAGCT | ACAAGCATAAATGTGTCAGCCTCTT |
| AIM2 | ATCTAGGCTGATCCTGGGACTGT | GTCCAGGCCGGTCAACAAC |

(clone 30-F11; BioLegend), anti-E-cadherin (Cat. no. 147303; Bio-Legend), anti-CD3 (clone 145-2C11; Invitrogen), anti-IgM (clone EB121-15F9; Invitrogen), anti-Ly6G (clone HK1.4; BioLegend), and anti-F4/80 (clone BM8; BioLegend). Flow cytometry was performed using hierarchical gating on an FACSARIA III (BD Biosciences), and RNA was isolated from sorted epithelial cells by Solution D.

### Polysome profiling

Polysomes were prepared as previously described (Nagarajan & Grewal, 2014). Briefly, the entire mouse kidney and intestine (small and large) were rinsed with cold saline containing cycloheximide 100 μg/ml and homogenized with a mini-dounce homogenizer in freshly prepared lysis buffer containing 1% (vol/vol) Triton X-100, 0.5% (wt/vol) sodium deoxycholate, 0.5 mM DTT, 100 μg/ml cycloheximide, 1 mg/ml heparin, 1× Roche mini protease inhibitor tab, 2.5 μM PMSF, 2.5 mM sodium fluoride, 1 mM sodium orthovanadate, 100 U/ml Riboblock (Fermentas) all in 25 mM Tris pH 7.4, 10 mM magnesium chloride, and 250 mM sodium chloride. Samples were processed with 10 passes through a 25G needle, cleared at 12,000 g, and the supernatant layered on a 12 ml 15–45% sucrose gradient for ultracentrifugation at 37,000 rpm (SW41 Beckman rotor, Optima L-90-K; Beckman Coulter) for 4°C, 2 h 30 mins. Fractionation was carried out on the BR188 Density Gradient Fractionation System (Brandel) with 20 fractions collected per sample with continuous spectrophotometric absorbance measurements at 254 nM. RNA was then precipitated overnight in 1:50 volume 5 M sodium chloride and 2.5 volumes 100% ethanol at –20°C, isolated by Solution D method, and further purified by precipitation with 7.5 M lithium chloride before heparinase treatment (0.4 U/ml, 1 h at room temperature) and cDNA synthesis.

### Nuclear/cytoplasmic fractionation

Whole mouse kidneys and intestine were resected, rinsed in cold saline, and 20 mg distributed for homogenization in a dounce homogenizer in 500 μl lysis buffer containing 50 mM Tris pH 8.0, 140 mM sodium chloride, 1.5 mM magnesium chloride, 0.5% (vol/vol) NP-40, 1 mM DTT, 0.2 U/uL RNase OUT. Nuclei were pelleted at 300g for 2 min, and RNA from the supernatant and nuclear fractions was extracted by Solution D.

### Cloning of NLRP6 leaders and luciferase chimeras

Chimeric 5′UTR-luciferase constructs were cloned by FastCloning using Phusion HF polymerase according to manufacturer's protocol (NEB) (Li et al, 2011). pGEM-luc and pSP64 Poly(A) vectors were purchased from Promega. NotI and BglII sites were first cloned into pSP64 Poly(A) vector, and luciferase was cloned upstream of the poly(A) sequence. 5′UTR from intestine and kidney/liver isoforms were cloned from GeneBlocks (Integrated DNA Technologies) directly 5′ to the luciferase sequence. Plasmid sequences were all verified by Sanger sequencing across inserts.

### In vitro transcription and translation

Each DNA construct was linearized by BglII digestion, and 1 μg of final phenol–chloroform extracted/ethanol precipitated DNA was used for input to the mMessage mMachine SP6 transcription kit (Thermo Fisher Scientific). Synthetic RNA was purified by phenol–chloroform extraction, and sample integrity and concentration verified by an Agilent Tapestation. RNA samples were aliquoted and maintained at −80°C. For in vitro translation, rabbit reticulocyte IVT lysate (Thermo Fisher Scientific) was used according to the manufacturers' protocol and input RNA concentration empirically determined. Luciferase was then measured on a Monolight 3010 Luminometer (BD Biosciences) using the Luciferase Assay Detection System (Promega).

### Mouse studies

$Nlrp6^{+/+}$ and $Nlrp6^{-/-}$ mice (Millenium Pharmaceuticals Inc.) on a C57Bl/6 background were derived from $Nlrp6^{-/+}$ breeding pairs (Chen et al, 2011). Mice used in these studies were F1 littermates. The UUO model was performed as previously described (Vilaysane et al, 2010). Mice were sacrificed at 7 and 14 d for analysis. Anti-GBM glomerulonephritis was induced in mice using heat-inactivated sheep anti-GBM nephrotoxic serum (NTS, generously provided by Dr L Mesnard) administered intravenously as previously described (Mesnard et al, 2009). Urine samples were collected on days 0, 4, and 9 following NTS injection and assayed for total protein (Bradford assay), albumin (Bethyl Laboratories), and creatinine (Exocel) as per the manufacturer's instruction. Mice were sacrificed at 9 d following NTS administration and kidneys collected for analysis.

### Statistics

Statistical analysis was done using both GraphPad Prism 8 and R version 3.3.3 (R Core Team, 2020). Where appropriate, ANOVA with Tukey's multiple comparison testing were used, with a $P$-value significance threshold of 0.05.

### Ethics

Human intestinal biopsies and human nephrectomy sample collection protocols were approved by the Conjoint Health Research Ethics Board at the University of Calgary and Alberta Health Services. All mouse studies were approved and conducted in accordance with guidelines set forth by the Animal Care Committee at the University of Calgary and conform to the guidelines set by the Canadian Council of Animal Care.

## Data Availability

Nlrp6+/+ and Nlrp6−/− RNA-Seq data are registered under NCBI BioProject PRJNA684477 and can be accessed using the following link: http://www.ncbi.nlm.nih.gov/bioproject/684477.

## Supplementary Information

## Acknowledgements

The authors thank Mona Chappellaz for her technical support. Infrastructure and technical support were also provided by the Flow Cytometry Facility, the Biobank for the Molecular Classification of Kidney Disease, and the Centre for Health Genomics and Informatics at the Snyder Institute for Chronic Diseases and the University of Calgary. Grant Support: This work was by supported by operating grants from the Canada Institutes for Health Research and the Kidney Foundation of Canada. Support was also provided by a team grant under the Canadian Institutes for Health Research (CIHR) Inflammation in Chronic Disease Signature Initiative.

### Author Contributions

NA Bracey: conceptualization, data curation, formal analysis, investigation, methodology, and writing—original draft, review, and editing.
JM Platnich: conceptualization, data curation, investigation, methodology, and writing—original draft.
A Lau: conceptualization, data curation, investigation, and methodology.
H Chung: data curation, investigation, and methodology.
ME Hyndman: resources and writing—review and editing.
JA MacDonald: resources, methodology, and writing—review and editing.
J Chun: conceptualization, supervision, methodology, and writing—review and editing.
PL Beck: resources.
SE Girardin: conceptualization and writing—review and editing.
PMK Gordon: conceptualization, data curation, formal analysis, investigation, methodology, and writing—original draft, review, and editing.
DA Muruve: conceptualization, resources, data curation, formal analysis, supervision, funding acquisition, validation, project administration, and writing—original draft, review, and editing.

### Conflict of Interest Statement

The authors declare that they have no conflict of interest.

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
