## [Reviewer comments · Life Science Alliance]

Life Science Alliance

Tissue-selective alternate promoters guide NLRP6 expression

Nathan Bracey, Jaye Platnich, Arthur Lau, Hyunjae Chung, Matthew Hyndman, Justin MacDonald, Justin Chun, Paul Beck, Stephen Girardin, Paul Gordon, and Daniel Muruve

DOI: <https://doi.org/10.26508/lsa.202000897>

Corresponding author(s): Daniel Muruve, University of Calgary

Review Timeline:	Submission Date:	2020-08-28
	Editorial Decision:	2020-11-17
	Revision Received:	2020-12-12
	Accepted:	2020-12-15

Scientific Editor: Shachi Bhatt

Transaction Report:

November 17, 2020

RE: Life Science Alliance Manuscript #LSA-2020-00897-T

Dr. Daniel Abraham Muruve
University of Calgary
Medicine
3280 Hospital Drive NW
Calgary, AB T2K 3M4
Canada

Dear Dr. Muruve,

Thank you for submitting your revised manuscript entitled "Tissue-selective alternate promoters guide NLRP6 expression".

As you will see from the appended reviewer reports, the referees are interested in the findings shown in the manuscript, and have suggested only minor edits for revision. We highly encourage you to submit a revised version to Life Science Alliance, and would be happy to publish your paper in Life Science Alliance pending minor revisions that address the reviewers' concerns.

Along with the points listed below, we would also encourage you to attend to the following,

- please add Keywords and Category to the system
- please add Author Contributions to our system
- please add a Summary Blurb / Alternate Abstract to the system
- please upload your main and supplementary figures as single files
- please use the [10 author names, et al.] format in your references (i.e. limit the author names to the first 10)
- please add your supplementary figure & table legends to the main manuscript text
- please upload your tables in editable doc or excel format
- please add a callout for Table 5 in your main manuscript
- please add scale bars to figure 7C and 7H
- please deposit the big datasets (RNA seq etc.) in one of the relevant public databases and share the accession number in the manuscript under a 'Data Availability' section (<https://www.life-science-alliance.org/manuscript-prep#datadepot>)
- please provide the source images (unedited un-cropped gels) for Figure S2A and B
- please edit the legend for figure S5 clarifying that the two blots for Nlrp6 in the figure are in fact the same blot at different exposures
- please provide a point-by-point response to the reviewers' comments

A. FINAL FILES:

B. MANUSCRIPT ORGANIZATION AND FORMATTING:

Sincerely,

Shachi Bhatt, Ph.D.
Executive Editor
Life Science Alliance
<https://www.lsajournal.org/>
Tweet @SciBhatt @LSAJournal

Reviewer #2 (Comments to the Authors (Required)):

This manuscript by Bracey et al. nicely demonstrates that both human and mouse Nlrp6 genes use alternate promoters leading to different mRNA transcripts of these genes. They also show that the protein-encoding transcript is only present in the intestine, while transcripts using alternate promoters are present in other tissues but are not translated. The authors explain the translational repression of these transcripts by showing that they are retained in the nucleus. In addition, the authors assessed whether the noncoding Nlrp6 transcripts in mice regulate other gene expression by performing RNAseq analysis in kidneys of Nlrp6^{+/+} vs ^{-/-} littermates, which showed that only the 2 genes immediately upstream of the Nlrp6 gene were altered, presumably due to local chromatin rearrangements. In conclusion, this study significantly adds to our understanding of how tissue-specific expression of Nlrp6 is regulated. Moreover, by clearly showing that Nlrp6 protein is not expressed in mouse kidney or BMDMs, this study helps to reconcile other studies that have demonstrated phenotypes resulting from Nlrp6 deficiency in kidney and BMDMs. In fact, the authors also perform kidney injury experiments that - in accordance with the lack of Nlrp6 expression in the kidney - do not show a function for Nlrp6 in these models, in contrast to a recent study that showed a protected role for Nlrp6 during kidney injury.

Overall, the results presented are convincing and are a good advance to the field. I only have minor suggestions for improvement.

1. For their mouse studies the authors used Nlrp6^{-/-} mice originally described by Chen et al. in 2011. These Nlrp6^{-/-} mice were made in ES cells from 129 mice and then backcrossed to C57Bl/6. Can the authors please specify the specific background of these mice? Were they backcrossed to C57Bl/6J or to C57Bl/6N? Strain-dependent genomic differences are very common in laboratory mice. It would therefore be informative if the authors could specify the strain, and if they could profile pure C57Bl/6J and C57Bl/6N mice from commercial vendors (since these are most the commonly used mouse strains) for the use of the alternate Nlrp6 TSS in kidneys. This could also help to explain findings in studies that do not use littermate controls. It is possible that depending on using either C57Bl/6J or C57Bl/6N non-littermate controls affects the expression of Nlrp6.

2. Related to the previous point, I would ask the authors to discuss the discrepancy between their kidney injury findings and the findings by Valino-Rivas et al. a bit more profoundly. The authors state in the discussion 'Prior animal studies have found that Nlrp6 deletion results in more severe injury in a chemical-induced acute kidney injury mouse model, although Nlrp6 protein expression was not confirmed in this study⁴³'. This is not true: ref 43 did confirm Nlrp6 protein expression (see fig 1d and 1e in this paper), although ref 43 did not show Nlrp6KO controls for these analyses. Ref

43 did not use littermate controls but did not mention whether these controls were C57Bl/6N or J. In addition, the kidney injury model used was not the same as the one used by the authors. Can the authors please discuss this a bit more in detail?

3. From the figure legend, the difference between the left and the right graph of Fig 1A is not clear. Please specify.

4. Fig 1C is not clear. It would be better to provide a table listing which PYD genes have how many TSS. From the graph in Fig 1C one cannot see that.

5. In the discussion the authors state 'We did not detect any differentially expressed genes in kidneys from littermate Nlrp6^{+/+} and Nlrp6^{-/-} animals at baseline'. However, in Fig 7 they show that Ifitm2 and Pggghg were upregulated in kidneys of Nlrp6^{-/-} mice. Can the authors briefly discuss the potential implications of the alterations in these gene expressions? Do these genes have functions in inflammation that for instance could explain studies observing effects of Nlrp6 deficiency in BMDMs although BMDMs do not express Nlrp6?

6. In the discussion the authors only cite ref 44 when referring to littermate control studies that could not confirm the role of Nlrp6 in regulating the microbiota. In fact, there were 2 studies that confirmed this (Lemire et al 2017 and Mamantopoulos et al 2017). It would be fair to cite both of these studies.

7. Type in Figure legend 3B: NLRRP6

8. Fig legend 4C mentions a gating plot for CD45-Ecadherin⁺ epithelial cells but that plot is not present in the figure. In addition, the color codes in these gating plots of Fig 4C are not clear.

Reviewer #3 (Comments to the Authors (Required)):

In the MS the authors describe a new mechanism by which the tissue expression of pyrin domain containing proteins are regulated predominantly using NLRP6 as their model. As discussed with the editor I am not qualified to comment on the quality of the molecular studies explored in this MS. The authors propose different mechanisms by which mouse and human NLRP6, particularly translationally, is regulated in tissues. This is interesting because the species differences between human and mice in NLRs are quite different and this study introduces another level of complexity to this area of research. This is important because many conclusions are extrapolated from mice to humans and it can be very misleading which is concerning where disease driven studies or decisions on new therapeutic targets are being made. Specific comments on this MS will come from the molecular referees, but this study contains interesting data which will be interesting for the scientific community.

Thank you for the thoughtful review of our manuscript. We have addressed all reviewer comments and adjusted the manuscript accordingly.

Reviewer 2

- 1. For their mouse studies the authors used Nlrp6^{-/-} mice originally described by Chen et al. in 2011. These Nlrp6^{-/-} mice were made in ES cells from 129 mice and then backcrossed to C57Bl/6. Can the authors please specify the specific background of these mice? Were they backcrossed to C57Bl/6J or to C57Bl6/N? Strain-dependent genomic differences are very common in laboratory mice. It would therefore be informative if the authors could specify the strain, and if they could profile pure C57Bl/6J and C57Bl/6N mice from commercial vendors (since these are most the commonly used mouse strains) for the use of the alternate Nlrp6 TSS in kidneys. This could also help to explain findings in studies that do not use littermate controls. It is possible that depending on using either C57Bl/6J or C57Bl6/N non-littermate controls affects the expression of Nlrp6.***

We were unable to obtain previous data on the specific C57Bl/6 sub-strain (C57Bl/6J vs 6N) used to generate and subsequently breed these mice. However, the question of generalizability of the Nlrp6 Δ 5'UTR is important. To address this, we instead looked at completely different strains of mice. There was a similar pattern of expression of the Nlrp6 Δ 5'UTR in the liver and kidneys between our C57Bl/6 animals, Balb/c and 129 mice. Interestingly, BALB/c mice did not express substantial Nlrp6 in colon (similar to human) (Supplementary Figure 4B, C). Nevertheless, we did not find any expression of Nlrp6 Δ 5'UTR in intestinal tissue from all three strains, consistent with a generalizable mechanism for translational Nlrp6 gene silencing outside of the intestine.

- 2. Related to the previous point, I would ask the authors to discuss the discrepancy between their kidney injury findings and the findings by Valino-Rivas et al. a bit more profoundly. The authors state in the discussion 'Prior animal studies have found that Nlrp6 deletion results in more severe injury in a chemical-induced acute kidney injury mouse model, although Nlrp6 protein expression was not confirmed in this study⁴³'. This is not true: ref 43 did confirm Nlrp6 protein expression (see fig 1d and 1e in this paper), although ref 43 did not show Nlrp6KO controls for these analyses. Ref 43 did not use littermate controls but did not mention whether these controls were C57Bl/6N or J. In addition, the kidney injury model used was not the same as the one used by the authors. Can the authors please discuss this a bit more in detail?***

We have edited the discussion to include a section addressing the discrepancy between our findings and those of Valino-Rivas et al. Primarily, we clarified that Nlrp6 expression in that study was not confirmed using Nlrp6^{-/-} negative controls raising the possibility of non-specificity in their Nlrp6 protein analysis. We altered the language in the discussion around the use of littermates and other potential mechanisms that might explain the discrepancy between the two studies.

- 3. From the figure legend, the difference between the left and the right graph of Fig 1A is not clear. Please specify.**

We clarified in the legend and in text that the left figure is the actual domain-encoding nucleotide length, whereas the right is the entire exon encoding that domain. Looking at both separately has implications for identifying relationships between PYD-containing genes (and indeed they clustered into distinct groups).

- 4. Fig 1C is not clear. It would be better to provide a table listing which PYD genes have how many TSS. From the graph in Fig 1C one cannot see that.**

We included a supplementary table to show the data more transparently.

- 5. In the discussion the authors state 'We did not detect any differentially expressed genes in kidneys from littermate Nlrp6+/+ and Nlrp6-/- animals at baseline'. However, in Fig 7 they show that Ifitm2 and Pgghg were upregulated in kidneys of Nlrp6-/- mice. Can the authors briefly discuss the potential implications of the alterations in these gene expressions? Do these genes have functions in inflammation that for instance could explain studies observing effects of Nlrp6 deficiency in BMDMs although BMDMs do not express Nlrp6?**

We revised the language regarding the differential gene expression in the RNASeq study. We also changed the discussion to include a reference for IFITM2 and raised the possibility that the phenotype in BMDM previously attributed to Nlrp6 could relate to off target effects.

- 6. In the discussion the authors only cite ref 44 when referring to littermate control studies that could not confirm the role of Nlrp6 in regulating the microbiota. In fact, there were 2 studies that confirmed this (Lemire et al 2017 and Mamantopoulos et al 2017). It would be fair to cite both of these studies.**

We agree and added the 2nd citation.

- 7. Type in Figure legend 3B: NLRRP6**

We corrected the typing error.

- 8. Fig legend 4C mentions a gating plot for CD45-Ecadherin+ epithelial cells but that plot is not present in the figure. In addition, the color codes in these gating plots of Fig 4C are not clear.**

We modified the figure to include the gating for E-cadherin and CD45.

We thank reviewer 3 for their positive comments.

Editorial Issues

- We added keywords and categories to the system
- We added author contributions to the system
- We added the blurb
- All figures are uploaded individually
- We reformatted the references
- All legends are now in the manuscript file
- We uploaded the tables as editable files
- We placed a call out to Table 5 in the methods section
- Scale bars are in figures 7
- We have uploaded RNASeq data to public databases and marked this in the manuscript
- We included all source images
- We edited all legends to clarify when exposures are done on the same membrane

December 15, 2020

RE: Life Science Alliance Manuscript #LSA-2020-00897-TR

Dr. Daniel Abraham Muruve
University of Calgary
Medicine
3280 Hospital Drive NW
Calgary, AB T2K 3M4
Canada

Dear Dr. Muruve,

Thank you for submitting your Research Article entitled "Tissue-selective alternate promoters guide NLRP6 expression". It is a pleasure to let you know that your manuscript is now accepted for publication in Life Science Alliance. Congratulations on this interesting work.

We noticed that Figure S2 includes both high and low exposures of the same membrane and have requested our typesetters to add the following to the legend of Figure S2:
"Both high and low exposures of the same membrane are shown."
Please let us know immediately if this is not correct.

*****IMPORTANT:** If you will be unreachable at any time, please provide us with the email address of an alternate author. Failure to respond to routine queries may lead to unavoidable delays in publication.*******

DISTRIBUTION OF MATERIALS:

Again, congratulations on a very nice paper. I hope you found the review process to be constructive and are pleased with how the manuscript was handled editorially. We look forward to future exciting submissions from your lab.

Sincerely,

Shachi Bhatt, Ph.D.

Executive Editor

Life Science Alliance

<https://www.lsajournal.org/>
